# Architectural Inductive Biases Can Be Enough for State Abstraction in Deep Reinforcement Learning

## Abstract

The ability to ignore task irrelevant environment variables is central to intelligent behavior. In reinforcement learning (RL), existing methods typically rely on auxiliary objectives to learn similar forms of abstraction. Such objectives tend to add significant complexity to the base RL algorithm. In this work, we take a step back and ask: can selective abstraction emerge naturally from reward optimization alone, without any additional objectives? Following prior work, we show that standard deep RL learns slowly or not at all in the presence of distracting, task-irrelevant state variables, failing to learn meaningful state abstractions. We then introduce a surprisingly simple neural network architecture change: a learnable, observation-independent attention mask applied to the inputs of the policy and value networks and trained end-to-end using only the RL objective. Despite its simplicity, this architectural modification consistently improves sample efficiency and learns to mask out distracting input variables across 12 continuous control tasks. We analyze the dynamics of gradient descent using this method on a linear regression task and demonstrate improved feature credit assignment. Finally, we conduct experiments on toy MDPs and show that the attention mask leads to accurate Q-value estimation and induces soft abstractions over a factored state space. Our findings challenge the need for complex auxiliary objectives to learn state abstractions in deep RL and suggest a simple baseline for future research.

## 1 Introduction

In complex environments, reinforcement learning (RL) agents observe a wide range of environment variables of which only a subset are relevant for decision-making. For example, a household robot may simultaneously perceive furniture, humans, ambient sound levels, lighting conditions, temperature, and floor texture, yet only certain features contribute meaningfully to its task at hand. This abundance of candidate features is a challenge for RL algorithms without prior knowledge of which variables are relevant. RL algorithms without such prior knowledge can learn policies and value functions that incorrectly depend upon task irrelevant variables and consequently fail to learn efficiently (Wang et al., 2024).

In order to identify and discard task-irrelevant state variables, prior work has introduced various auxiliary objectives to provide learning signals beyond the reward alone. These include metrics that quantify state similarity based on reward and transition dynamics (Zhang et al., 2020; Castro et al., 2021) and learning through causal reward modeling (Wang et al., 2024). However, the role of architectural choices in enabling or shaping state abstraction has received comparatively little attention. Motivated by this gap, we aim to answer the following question:

*Are architectural choices and the RL objective alone sufficient to learn abstract state representations?*

In this work, we show that the answer can be yes. Building on the framework of Wang et al. (2024), we investigate the capacity of neural networks to suppress distracting input variables. We first reaffirm that standard architectures used for continuous control benchmarks, such as multi-layered perceptrons (MLPs), learn slower in the presence of such distractors, as prior work has also shown (Wang et al., 2024).

We then introduce a learnable attention mask applied to the inputs of the neural network. Our approach draws inspiration from prior work on masking mechanisms, which have been integrated into various architectures and shaped using diverse optimization objectives (Wang et al., 2024; Wu et al., 2021; Grooten et al., 2023; Salter et al., 2021). However, a key distinction lies in the nature of the masking: existing methods are typically context-dependent, suppressing features only locally when it negatively affects performance, whereas we use a simpler observation-independent mask. Moreover, there has been limited empirical or theoretical inquiry into why distractors are so detrimental or how such seemingly minor architectural modifications can produce substantial gains in performance. Saxe et al. (2019) show that MLPs experience a loss in mutual information between distracting inputs and hidden layers of an MLP, but not enough to consistently recover optimal performance in MuJoCo control tasks, as we demonstrate. We provide insight into this phenomenon from two complementary perspectives. First, we perform a gradient dynamics analysis of stochastic gradient descent updates in-expectation for linear models with and without attention-based masking. We show that the detrimental effects of distractors and the benefits of bounded masking extend beyond RL to general function approximation. Second, using a Deep Q-Network (DQN) (Mnih et al., 2015) trained on randomly generated toy MDPs adapted from Yang et al. (2022), we show that even in non-linear regimes, our method yields statistically significantly better estimates of the optimal Q-value function compared to vanilla MLPs.

Together, our findings show that an architecture inductive bias, driven solely by the reward signal, is sufficient to give rise to an abstract state representation. These findings question the need for auxiliary losses to induce appropriate state abstraction.

## 2 RELATED WORK

In this section, we review the significant literature on learning state abstractions, abstract state representations, and handling distracting inputs in deep RL.

### 2.1 LEARNING APPROXIMATE STATE ABSTRACTIONS

Bisimulation (Givan et al., 2003) formalizes exact abstraction by grouping states that have behaviorally indistinguishable dynamics. In contrast, MDP homomorphisms (Ravindran, 2004) define a more flexible surjective mapping from ground to abstract states that preserves rewards and transition structure in expectation, allowing dissimilar states to be merged as long as abstract behavior remains approximately faithful. In subsequent work, Dean et al. (2013) relaxed the exact equivalence of bisimulations to aggregate states that behave approximately the same in a factored representation of a bounded parameter MDP. Taylor et al. (2008) relate MDP homomorphisms with lax bisimulation and devise a metric on states to provide approximation guarantees. Abel et al. (2016) show that the error in behavior due to approximate abstractions is polynomially bounded, while approximation does not require solving the exact MDP and allows for a greater degree of compression and tunable strictness of abstraction. Li et al. (2006) provide a unifying framework for different types of abstractions and show that under certain conditions, approximate abstractions can still lead to near-optimal policies, motivating the study of lossy but useful state representations. We extend this line of work by showing that graded abstractions can be induced implicitly through end-to-end learning in RL. Auxiliary losses based on bisimulation metrics (Ferns et al., 2004) have been introduced to shape the feature space such that the distance between two states' representations reflects their behavioral similarity in the MDP (Zhang et al., 2020; Castro et al., 2021). Other approaches such as DeepMDP (Gelada et al., 2019), learn a latent MDP model by predicting both rewards and next-state distributions in latent space. These objectives ensure that states with dissimilar transitions or rewards are embedded distinctly, thereby preserving bisimulation-based structure. In contrast, we show that an appropriately biased architecture can eliminate the need for such auxiliary supervision, learning task-aligned abstractions solely through interaction and reward feedback.

### 2.2 HANDLING DISTRACTIONS IN DEEP RL

A common approach to studying distractors in RL involves appending irrelevant variables to state observations. Our work builds on the setup of Wang et al. (2024), who learn a binary mask via causal and reward models to identify variables that influence dynamics or reward, yielding bisimulation-

consistent abstractions. Much of the prior work on distractors focuses on visual domains. The DMC Distracting Control Suite (Stone et al., 2021) adds noise through camera variation and background motion. In this setting, Zhou et al. (2023) use sequential reward prediction to shape representations, while Liu et al. (2023) use bisimulation distances and prototype clustering for robustness. Our work is related to approaches that leverage architectural attention: Bramlage and Cortese (2022) incorporate self-attention into policy and value networks; Mott et al. (2019) introduce a recurrent attention model using key-query-value attention (Vaswani et al., 2017); and Salter et al. (2021) show its benefits in noisy visual tasks. Some methods fully decouple distractor suppression from policy learning: Wang et al. (2021) extract invariant foreground features via keypoint detection, while Wu et al. (2021) learn input-dependent attention masks through reconstruction, though these are task-agnostic. Most recently, Grooten et al. (2023) train an observation-conditioned mask using only critic loss. In contrast to these methods, our approach is solely reward-guided and *observation-independent*, enabling it to identify globally relevant features across the entire observation space, those most predictive of returns.

## 3 PROBLEM SETTING: FACTORED MDPS WITH TASK-IRRELEVANT DISTRACTORS

We consider the setup introduced in Wang et al. (2024), where observations are state-based and contain both task-relevant and task-irrelevant components. We model these environments as *factored MDPs*, defined by the tuple $\mathcal{M} = (\mathbb{S}, \mathbb{A}, P, r, \gamma)$, where $s \in \mathbb{S}$ denotes the state, $a \in \mathbb{A}$ the action, $P : \mathbb{S} \times \mathbb{A} \times \mathbb{S} \to [0, 1]$ is the transition probability function with $P(s' \mid s, a) = \mathbb{P}(s_{t+1} = s' \mid s_t = s, a_t = a)$, $r : \mathbb{S} \times \mathbb{A} \to \mathbb{R}$ is the reward function, and $\gamma \in [0, 1)$ is the discount factor. A stochastic policy $\pi : \mathbb{S} \times \mathbb{A} \to [0, 1]$ defines a distribution over actions, such that $\pi(s, a) = \mathbb{P}(a_t = a \mid s_t = s)$.

The state space factorizes as $\mathbb{S} = \mathbb{X}_{\text{rel}} \times \mathbb{X}_{\text{irr}}$. A full state $s \in \mathbb{S}$ is represented as $s = (x_{\text{rel}}, x_{\text{irr}})$, where $x_{\text{rel}} \in \mathbb{X}_{\text{rel}}$ comprises task-relevant variables that influence both the transition dynamics and the reward, and $x_{\text{irr}} \in \mathbb{X}_{\text{irr}}$ comprises task-irrelevant distractors that evolve independently and have no causal influence on either reward or dynamics. Specifically, the transition and reward functions factor as:

$$P(s' \mid s, a) = P(x'_{\text{rel}} \mid x_{\text{rel}}, a) \cdot P(x'_{\text{irr}} \mid x_{\text{irr}}, a), \quad r(s, a) = r(x_{\text{rel}}, a). \tag{1}$$

In practice, the exact factorization may only hold approximately. In our experiments, we include a mixture of distractor types: some evolve independently of the agent's actions (e.g., randomly sampled or following their own stochastic processes), while others evolve conditionally on the agent's actions via $P(x'_{\text{irr}} \mid x_{\text{irr}}, a)$. Crucially, these variables remain irrelevant to both the reward and the transitions of $x_{\text{rel}}$, thereby they are unnecessary inputs for the optimal policy. This relaxation from strict independence allows us to simulate more realistic distractor dynamics.

## 4 STATE ABSTRACTION THROUGH OBSERVATION-INDEPENDENT INPUT MASKING

Given a standard model-free deep reinforcement learning (RL) algorithm (e.g., SAC Haarnoja et al. (2018)), we introduce a lightweight architectural module designed to identify and discard task-irrelevant variables induce task-specific abstraction throughout end-to-end training. Let the observation space be factored as $s = (x_1, x_2, \ldots, x_n) \in \mathbb{R}^n$, where each $x_i$ represents an individual state variable. We associate with each variable $x_i$ a corresponding learnable parameter $\phi_i \in \mathbb{R}$, and collectively define the masking parameter vector $\phi = (\phi_1, \ldots, \phi_n) \in \mathbb{R}^n$. We initialize these parameters to zero and share the same mask across all function approximators involved (e.g., the policy and value networks).

At each training step, we compute a gating vector $\alpha = \sigma(\phi) \in (0, 1)^n$, where $\sigma(\cdot)$ is the element-wise sigmoid activation, and apply it to the input via a Hadamard product: $\tilde{s} = \alpha \odot s$. This masked observation is used as input to both the actor and critic networks.

For actor-critic methods, we found it beneficial to update $\phi$ using either the actor or critic loss—but not both. To enforce this, we stop gradients from flowing through the loss not being used. For instance, if updating via the actor loss only, the critic receives $\tilde{s}_{\text{critic}} = \texttt{detach}(\alpha) \odot s$, while the

actor receives $\tilde{s}_{\text{actor}} = \boldsymbol{\alpha} \odot \boldsymbol{s}$. The gating parameters are then updated via backpropagation through the actor loss:

$$\boldsymbol{\phi} \leftarrow \boldsymbol{\phi} - \eta_{\boldsymbol{\phi}} \nabla_{\boldsymbol{\phi}} \mathcal{L}_{\text{actor}}(\boldsymbol{\theta}, \boldsymbol{\phi}), \tag{2}$$

where $\mathcal{L}_{\text{actor}}(\boldsymbol{\theta}, \boldsymbol{\phi})$ denotes the actor loss (e.g., from PPO or SAC), computed with respect to the policy network parameters $\boldsymbol{\theta}$ and the shared masking parameters $\boldsymbol{\phi}$. The learning rate $\eta_{\boldsymbol{\phi}}$ controls the step size for updating $\boldsymbol{\phi}$. Note that while $\boldsymbol{\theta}$ governs the weights of the policy network, $\boldsymbol{\phi}$ is a separate parameter vector whose gradients are computed solely through its influence on the masked input $\tilde{s}_{\text{actor}}$.

This setup ensures that $\boldsymbol{\phi}$ is learned solely from the RL task objective, yielding a *soft* abstraction rather than a hard partition over the state space. Since the masking vector $\boldsymbol{\alpha} = \sigma(\boldsymbol{\phi})$ lies in $(0, 1)^n$, each variable is only partially suppressed, allowing a graded notion of relevance. This continuous relaxation enables differentiable credit assignment and supports gradient-based optimization. In the ideal case, the masking vector associates a weight of almost zero with all task-irrelevant inputs and a weight of almost one with task-relevant variables. In practice, this hard abstraction is not necessarily recovered but, as we will show, the weights still correctly suppress task-irrelevant variables more and doing this leads to faster learning in the presence of such variables.

## 5 DM CONTROL SUITE EXPERIMENTS

To evaluate the effectiveness of our proposed abstraction mechanism, we conduct extensive experiments across a range of continuous control benchmarks. We show that the presence of distractors in the state space leads to significant performance degradation of SAC, TD3, and PPO in standard MuJoCo tasks compared to learning without the distractors. We then show that the simple architecture modification we introduced in the previous section is sufficient to significantly decrease the gap between the methods. These results show that the RL objective and a simple architecture can be sufficient for learning in the presence of such distractors.

### 5.1 EMPIRICAL SETUP

**Tasks.** Our experiments span 12 continuous control tasks from the DeepMind Control Suite (MuJoCo): `walker-walk`, `walker-run`, `cheetah-run`, `hopper-hop`, `hopper-stand`, `finger-spin`, `finger-turn_easy`, `finger-turn_hard`, `fish-swim`, `fish-upright`, `reacher-hard`, and `swimmer-swimmer6`. These tasks cover a broad spectrum of locomotion and manipulation challenges, with varying levels of complexity in dynamics, control frequency, and reward structure. The diversity of tasks ensures that our findings are not tied to a narrow class of dynamics or reward functions. Each environment features continuous state and action spaces.

**Distractor Augmentation.** The distractor augmented MDP is instantiated using two types of distractor variables: *uncontrollable* and *controllable* (Wang et al., 2024). Let $\boldsymbol{a}_t \in \mathbb{R}^d$ denote the action taken at time step $t$. Uncontrollable distractors are modeled as noise vectors $\boldsymbol{x}_t^{(\text{unc})} \in \mathbb{R}^{d_{\text{unc}}}$, sampled independently at each time step:

$$\boldsymbol{x}_t^{(\text{unc})} \sim \mathcal{U}(\boldsymbol{\mu}_{\text{unc}} - \boldsymbol{\delta}, \boldsymbol{\mu}_{\text{unc}} + \boldsymbol{\delta}), \tag{3}$$

where $\boldsymbol{\mu}_{\text{unc}} \in \mathbb{R}^{d_{\text{unc}}}$ is a fixed bias sampled once at the beginning of each experiment, and $\boldsymbol{\delta} \in \mathbb{R}_+^{d_{\text{unc}}}$ defines the range of variation. These variables evolve independently of the agent's behavior and serve as purely exogenous noise.

Controllable distractors, by contrast, evolve deterministically as a function of the agent's actions. At each time step, they are generated by an affine transformation of the current action:

$$\boldsymbol{x}_t^{(\text{con})} = \boldsymbol{W} \boldsymbol{a}_t + \boldsymbol{b}, \tag{4}$$

where $\boldsymbol{W} \in \mathbb{R}^{d_{\text{con}} \times d}$ is a weight matrix and $\boldsymbol{b} \in \mathbb{R}^{d_{\text{con}}}$ is a bias vector. Both $\boldsymbol{W}$ and $\boldsymbol{b}$ are sampled uniformly at random once per experiment and held fixed throughout. This ensures that controllable distractors are correlated with the agent's behavior but remain irrelevant to task performance, as they are not part of the reward or transition-generating processes for $\boldsymbol{x}_{\text{rel}}$.

To simulate high-dimensional, distractor-laden observations, we augment the native task-relevant state vector $x_{\text{rel}}$ with 40 task-irrelevant variables, comprising 20 controllable and 20 uncontrollable distractor dimensions, resulting in a factored observation of the form $s = (x_{\text{rel}}, x_{\text{con}}, x_{\text{unc}})$. This allows us to test the agent's ability to filter out noise across both deterministic and stochastic distractor sources in a variety of control settings.

**Implementation Details.** We evaluate our learned masking mechanism across three deep reinforcement learning algorithms: Soft Actor-Critic (SAC) (Haarnoja et al., 2018), Twin Delayed Deep Deterministic Policy Gradient (TD3) (Fujimoto et al., 2018), and Proximal Policy Optimization (PPO)(Schulman et al., 2017). In all cases, the actor and critic networks are implemented as multilayer perceptrons (MLPs), with ReLU activations used for SAC and TD3, and Tanh activations for PPO. The temperature of the sigmoid is kept at 1 for these experiments. SAC and TD3 are trained off-policy using a replay buffer of size 1 million and run for 1 million environment steps per seed. For SAC, the entropy regularization coefficient is automatically tuned using a dedicated Adam optimizer. PPO, being an on-policy algorithm, is trained longer for 3 million steps. The full set of hyperparameters for each algorithm will be provided in the Appendix C. The results are compared against two baselines. First, *oracle* receives only task relevant variables as input (the ground truth abstraction) and thus serves as a strong upper bound on performance in that task and, second, *full* observes the full state augmented with distractors, which we show performs poorly compared to oracle performance.

All experiments were conducted on a high-throughput computing cluster. The compute pool consists of heterogeneous CPU-only worker nodes with Intel Xeon processors, ranging from 8 to 64 cores and 16–256 GB of RAM per node.

**Evaluation Protocol.** SAC and TD3 use a fixed protocol: after every training episode, the agent is evaluated on 50 episodic rollouts using a separate environment with independently sampled distractor biases and projection matrices. This ensures that the evaluation distractors are in-distribution but different from those seen during training. The average episodic return across the 50 test episodes is logged. For PPO, evaluation is conducted online by recording episodic returns directly during training rollouts.

## 5.2 RESULTS

Figure 1 reports the average performance of each algorithm across all 12 tasks. The baseline setting using vanilla MLP function approximators performs consistently worse than the oracle, particularly in the presence of distractors. In contrast, our method closes this gap, recovering performance towards the oracle despite having access only to reward-based supervision. Individual plots for each task are provided in the Appendix A and show, in many cases, the ability to match the oracle upper bound.

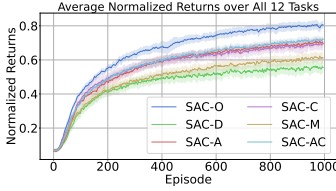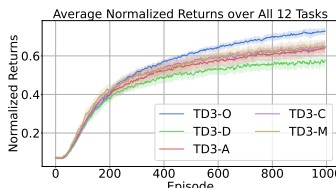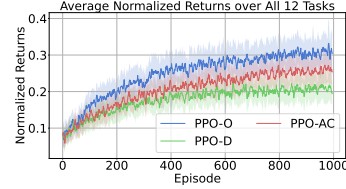

Figure 1: Performance of SAC, TD3, and PPO averaged across 12 MuJoCo tasks. For each of the algorithms—Oracle (–O), distractors without attention (–D), attention trained with the actor (–A), attention trained with the critic (–C), separate attention trained with both (–AC) and MaDi (–M)—the performance curves are computed as the mean across all tasks. These per-seed curves are then averaged across the 10 random seeds with 95% confidence intervals to obtain the final aggregated results as shown.

**Actor vs. Critic.** When using the learned attention mask, using either the actor or critic signal in isolation yields stable training, suggesting that each loss alone is sufficient to drive task-aligned abstraction. For SAC, we also evaluate a variant where separate masks are trained for the actor and critic using their respective gradients. All these approaches achieve comparable performance. This similarity may stem from an implicit overlap in relevant features: in many continuous control tasks, variables useful for value estimation also aid policy learning. In contrast, training a single

shared mask using both losses simultaneously results in degenerate attention weights and near-zero returns. This indicates that the actor and critic provide conflicting gradient signals when applied jointly, consistent with findings from Garcin et al. (2025), which show that actor and critic networks tend to learn representations that are optimized for different purposes.

Interestingly, we find that in PPO updating the masking parameters using both actor and critic losses yields performance indistinguishable from using either loss alone. This may be due to the shared backbone architecture of PPO and the synchronized updates of the actor and critic, which reduce the representational divergence between the two losses, minimizing the gradient conflict we observe in off-policy methods.

**Comparison with MaDi.** Our observation-independent masking performs comparably (for TD3) or better (for SAC) than the observation-conditioned mask proposed in Grooten et al. (2023). In contrast to our approach, such methods may overfit to idiosyncrasies in the input and learn contextual abstractions, rather than identifying a globally relevant set of features for RL.

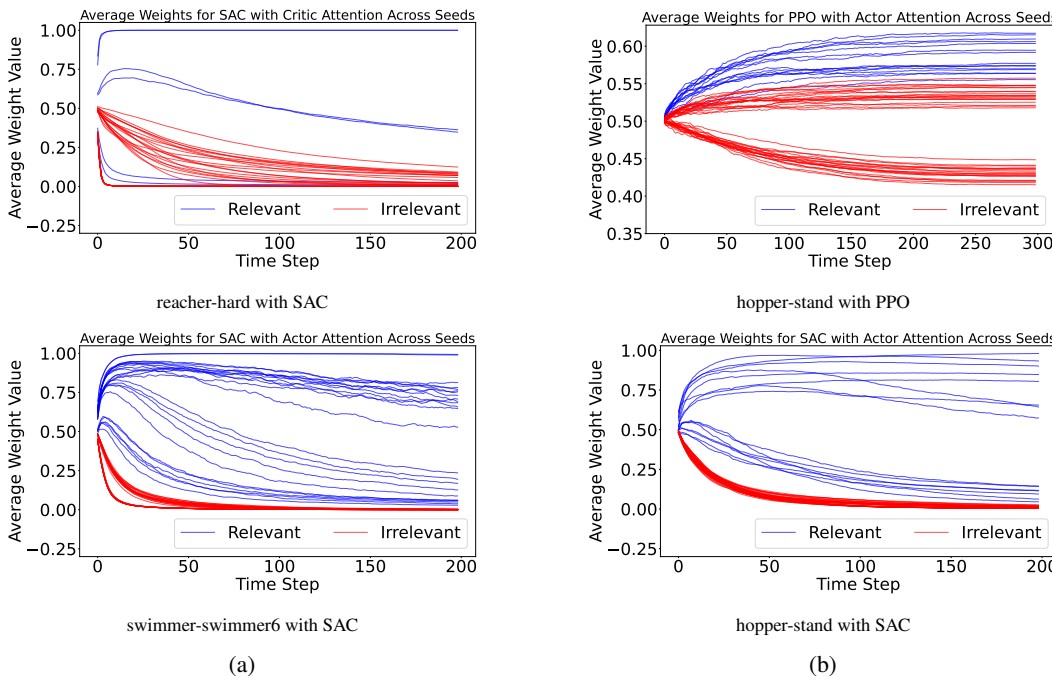

reacher-hard with SAC          hopper-stand with PPO

swimmer-swimmer6 with SAC          hopper-stand with SAC

(a)                      (b)

Figure 2: Trajectory of average attention weights across training. (a) SAC on `reacher-hard` and `swimmer-swimmer6`: confident masks emerge in easier tasks like `reacher-hard`, while `swimmer` shows more ambiguous gating. (b) PPO vs. SAC on `hopper-stand`: SAC shows sharper convergence; PPO remains diffuse. Task-relevant features are highlighted in blue, task-irrelevant in red. Results aggregated over 10 seeds.

**Correctness of Learned Abstraction.** We track the evolution of attention weights during training and plot the weights for each observation variable for two distinct environments using the SAC agent in Figure 2a. While the attention weights do not strictly go to either 0 or 1 (producing a hard abstraction), they generally correspond to task-relevant variables receiving higher weights and task-irrelevant variables receiving lower weights. Notably, we observe that the sharpness of the learned gating values and the overall task performance are correlated. In environments where the agent performs well, such as `reacher-hard`, the attention mechanism tends to converge to a confident binary-like mask. As shown in Figure 2a, most attention weights (all but two) converge to either 0 or 1, indicating that the model has learned to clearly distinguish between relevant and irrelevant input variables. For more challenging tasks such as `swimmer-swimmer6`, where the agent's performance is relatively lower, the learned attention masks are less confident. The attention weights remain diffuse and fail to clearly separate important features from distractors. However, while the mask is less sharp, it still preserves the correct relative ranking of feature relevance. Similar plots are provided for all environments in Appendix A.

As shown in Figure 2b, attention masks learned by on-policy methods like PPO fail to strongly suppress distractors, while off-policy methods such as SAC and TD3 learn sharper, more selective patterns that effectively mask out all task-irrelevant variables. A plausible explanation lies in differences in sample efficiency and data reuse (Queeney et al., 2021). On-policy methods discard trajectories after each update, limiting the diversity of experiences and restricting the refinement of internal representations like attention masks. In contrast, off-policy methods continually reuse past interactions, enabling more stable gradient estimates and better credit assignment, which in turn supports stronger attention patterns.

> **Takeaway #1**
>
> The attention mask we introduce and train solely with RL objectives is able to (1) down-weight the influence of distractors and (2) learn nearly as well as when the distractors are known.

## 6 CONTROLLED ANALYSIS

In this section, we show mathematically and through controlled toy experiments that an input-independent attention mask can lead to meaningful abstractions that discards task-irrelevant variables. Of particular note, we show mathematically that the presence of distractors can slow learning even when using *expected updates* (i.e., using infinite data to compute each gradient step). We then show that the input-independent attention mask serves to suppress the effect of the distracting variable.

### 6.1 GRADIENT DYNAMICS IN LINEAR MODELS

As critic training in RL amounts to repeatedly solving a regression task, in this section, we analyze the expected gradient updates for linear regression in the presence of distracting inputs. This setting is equivalent to learning a critic in a contextual bandit setting where value prediction reduces to supervised regression from state to reward. We analyze expected gradient descent updates for three settings: (i) in the absence of distractor variables (oracle case), (ii) in the presence of distractors without any form of masking or attention (full case), and (iii) in the presence of distractors with our proposed soft attention mechanism. We use expected updates (i.e., the infinite data regime) to focus on optimization rather than statistical challenges.

Consider a regression setting in which data is generated according to $Y = mX + c$ where $X \sim \mathcal{N}(0,1)$. There is also a distracting variable $D \sim \mathcal{U}(0,1)$. We consider training linear functions of the form $f_{\boldsymbol{w}}(x,d) = \boldsymbol{w}^\top [1, x, d]$ using gradient descent. The gradient descent update is

$$\boldsymbol{w}^{t+1} \leftarrow \boldsymbol{w}^t - \eta \, \mathbb{E}_{X \sim \mathcal{N}(0,1)} \mathbb{E}_{D \sim \mathcal{U}(0,1)} \left[ \nabla_{\boldsymbol{w}^t} (f_{\boldsymbol{w}^t}(X,D) - Y)^2 \right] \tag{5}$$

**Proposition 1** (Gradient Updates with Known Distractors (Oracle)). *The linear model is given as:*

$$f_{\boldsymbol{w}}(x) = w_0 + w_1 x \tag{6}$$

*and the expected update for $\boldsymbol{w}$ is given by:*

$$\langle w_0^{t+1}, w_1^{t+1} \rangle \leftarrow \langle (1-\eta)w_0^t + \eta c, \ (1-\eta)w_1^t + \eta m \rangle \tag{7}$$

**Proposition 2** (Gradient Updates with Unknown Distractors (Full)). *The linear model with distractors and no attention mechanism is given by:*

$$f_{\boldsymbol{w}}(x,d) = w_0 + w_1 x + w_2 d \tag{8}$$

*and the expected updates for each component of $\boldsymbol{w}$ are given by:*

$$w_0^{t+1} \leftarrow (1-\eta)w_0^t - \tfrac{\eta}{2} w_2^t + \eta c$$
$$w_1^{t+1} \leftarrow (1-\eta)w_1^t + \eta m$$
$$w_2^{t+1} \leftarrow \left(1 - \tfrac{\eta}{3}\right) w_2^t - \tfrac{\eta}{2} w_0^t + \tfrac{\eta}{2} c \tag{9}$$

**Observation 1** (Distracting inputs mislead bias learning in the full model). *With oracle knowledge of the distractor variable, the expected update results in $w_0 \to c$ and $w_1 \to m$ and the true data generating function is recovered. Without this oracle knowledge, the update for $w_1$ is unchanged but*

$w_0$ and $w_2$ *now depend upon one another and change to try and explain the bias term, c, in the data generation function. Consequently, for any non-zero values of* $w_0$ *and* $w_2$, $w_0$ *moves toward* $c - \frac{w_2}{2}$ *and* $w_2$ *moves toward* $\frac{3c}{2} - \frac{3w_0}{2}$, *albeit the latter moves at the slower rate of* $\frac{\eta}{3}$. *As the target for* $w_2$ *is only 0 when* $w_0 = 1$, *we do not, in general, expect* $w_2$ *to reach zero and d to be ignored.*

**Proposition 3** (Gradient Update: Attention-Based Model). *The linear model with input-independent attention mask is given as:*

$$f_{\boldsymbol{w}}(x, d) = w_0 + w_1(x\,\sigma(\phi_1)) + w_2(d\,\sigma(\phi_2)), \tag{10}$$

*where* $\phi_1$ *and* $\phi_2$ *are learnable parameters and* $\sigma$ *is the sigmoid function. The expected update equations for* $\boldsymbol{w}$, $\phi_1$, *and* $\phi_2$ *are given as:*

$$w_0^{t+1} \leftarrow w_0^t - \eta \left( w_0 + \frac{w_2\,\sigma(\phi_2)}{2} - c \right)$$

$$w_1^{t+1} \leftarrow w_1^t - \eta \left( w_1\,\sigma(\phi_1)^2 - m\,\sigma(\phi_1) \right)$$

$$w_2^{t+1} \leftarrow w_2^t - \eta\,\sigma(\phi_2) \left( \frac{w_0}{2} + \frac{w_2\,\sigma(\phi_2)}{3} - \frac{c}{2} \right)$$

$$\phi_1^{t+1} \leftarrow \phi_1^t - \eta\,w_1\,\sigma(\phi_1)(1 - \sigma(\phi_1))(w_1\,\sigma(\phi_1) - m)$$

$$\phi_2^{t+1} \leftarrow \phi_2^t - \eta\,w_2\,\sigma(\phi_2)(1 - \sigma(\phi_2)) \left( \frac{w_0}{2} + \frac{w_2\,\sigma(\phi_2)}{3} - \frac{c}{2} \right) \tag{11}$$

**Observation 2** (Attention mask updates suppress distractors.). *The direction of the update for* $\phi_2$ *is determined by the expression:*

$$w_2 \left( \frac{w_0}{2} + \frac{w_2\,\sigma(\phi_2)}{3} - \frac{c}{2} \right)$$

*as the factor* $\sigma(\phi_2)(1 - \sigma(\phi_2))$ *in Equation 11 is always non-negative and thus only serves to scale the update. When the distractor's contribution is large, i.e., when* $|w_2|$ *is large or* $\sigma(\phi_2)$ *closer to one, this term is typically positive, leading to a downward update that drives* $\sigma(\phi_2) \to 0$ *and effectively suppressing the distractor. Notably, suppression emerges without explicit knowledge of distracting variables; the model learns to attenuate irrelevant inputs purely from the error signal.*

**Empirical Validation**  We also validate these conclusions empirically in a synthetic regression setting using the derived gradient update equations for both the full and attention-based model. All weights are initialized using a uniform distribution $\mathcal{U}(-1/\sqrt{2}, 1/\sqrt{2})$, and updates are performed using fixed-step gradient descent with a step size of 0.01. For the attention-based model, the attention weights are initialized to zero and updated jointly with the main weights via backpropagation through the sigmoid gating functions. At each training step, we compute the loss over a sampled batch of 5,000 data points drawn from the true data-generating process. This process is repeated across 50 random seeds to account for variance due to initialization. As shown in Figure 3, even in the infinite data regime, the attention-based updates consistently converge faster and more stably to the optimal solution compared to updates without the attention interactions.

---

**Takeaway #2**

Distractors can lead to conflicting gradient updates even with infinite data; the attention mask is updated in a way that suppresses the distractors' influence.

---

### 6.2 POLICY EVALUATION IN TOY MDPS

Finally, we run a policy evaluation experiment in a controlled, low-dimensional MDP to understand if the attention mask is improving the accuracy of policy evaluation. The base environment is a custom MDP with continuous one-dimensional state space and two discrete actions adapted from Yang et al. (2022). The transition dynamics for each action are defined via randomly sampled piecewise linear functions, making the environment deterministic but non-trivial. We discretize the continuous state space into 20 evenly spaced states and compute the ground-truth Q-values via value iteration. For the full setting, the agent is trained with 20 additional task-irrelevant dimensions—10 controllable and 10 uncontrollable—appended to the 1D input. Each DQN variant is trained for 2,000 episodes using a small MLP. The distractors are generated similar to the main experiments. To evaluate

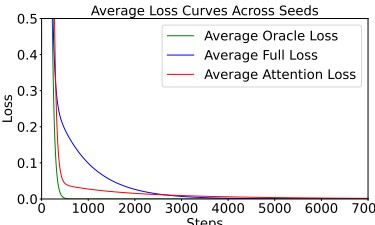 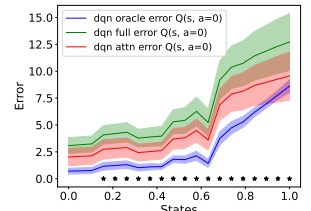 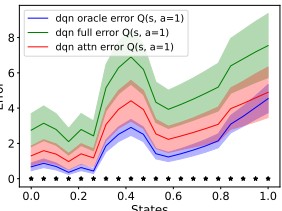

Figure 3: Loss with respect to the true data generating function. Results aggregated over 50 seeds.

Figure 4: Mean squared error of Q-value estimation for action 0 and 1 across oracle, full, and attention-based agents.

learned Q-values, we run all trained models on a fixed evaluation set of states with the same distractor statistics and compare the predicted Q-values against the ground-truth values for both actions using per-state mean squared error.

We repeat all experiments over 20 random seeds and report the mean Q-value estimation errors along with 95% confidence intervals. Additionally, we perform paired $t$-tests across seeds to assess the statistical significance of the differences in estimation error between the attention-based and vanilla models. Each black star denotes a statistically significant difference in Q-value estimation error for that state, as determined by a paired t-test. The results are presented in Figure 4 for 20 discretized states in a randomly generated MDP. The sigmoid temperature used for the attention masks in this setting is the default value of 1 as used in the main experiments. Plots for other temperature settings are provided in Appendix A. Temperature is a sensitive hyperparameter: higher values yield sharper, more confident masks, but excessively high temperatures can lead to vanishing gradients and unstable training, resulting in overly confident but incorrect masking. As shown, the attention-based model yields significantly lower errors than the baseline in most of the 20 discretized states, indicating more accurate approximation of the true Q-values.

> **Takeaway #3**
>
> The attention mechanism achieves statistically significant improvements in value estimation across the majority of states and all actions, even within the deep RL setting.

## 7 CONCLUSION

In this paper, we considered the problem of learning state abstractions in RL and asked the question of whether this was possible without relying on auxiliary objectives as done in prior work. We introduced a simple neural network architecture modification – an observation-independent attention mask applied to the inputs of the actor and critic networks – that is trained along with other network parameters using only RL objectives. Across 12 continuous control DMControl tasks augmented with task-irrelevant observation variables, we found that this small change enables agents to suppress task-irrelevant inputs and close the gap with agents with access to the ideal state abstraction. Through a combination of empirical evaluation and theoretical insights from linear settings, we demonstrate that this lightweight inductive bias supports selective credit assignment and facilitates the emergence of soft abstractions aligned with task dynamics and reward structure. Our findings suggest that, when combined with appropriate architectural constraints, the reward signal alone can suffice to induce abstraction, challenging the prevailing reliance on auxiliary losses in prior work. These findings (1) suggest that future work in abstraction learning should consider this simple architecture as a baseline before introducing more complex methods based on auxiliary losses and (2) open up a promising new research direction into designing architectures that promote abstraction through RL objectives alone.

### REPRODUCIBILITY STATEMENT

Full derivations for the results presented in Section 6 are provided in Appendix B. Plots for additional experiments can be found in Appendix A, and the complete code to reproduce all results in this paper

is available via an anonymous link in Appendix D. A detailed list of hyperparameters used for all experiments is included in Appendix C.

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

# A    ADDITIONAL RESULTS

(a) cheetah-run

(b) finger-spin

(c) finger-turneasy

(d) finger-turnhard

(e) fish-swim

(f) fish-upright

(g) hopper-hop

(h) hopper-stand

(i) reacher-hard

(j) swimmer-swimmer6

(k) walker-run

(l) walker-walk

Figure 5: Performance of SAC agents with and without the proposed observation-conditioned gating mechanism across various DM Control Suite tasks. The presence of distractors significantly impairs performance when no attention is applied. Incorporating our learned gating improves robustness and sample efficiency.

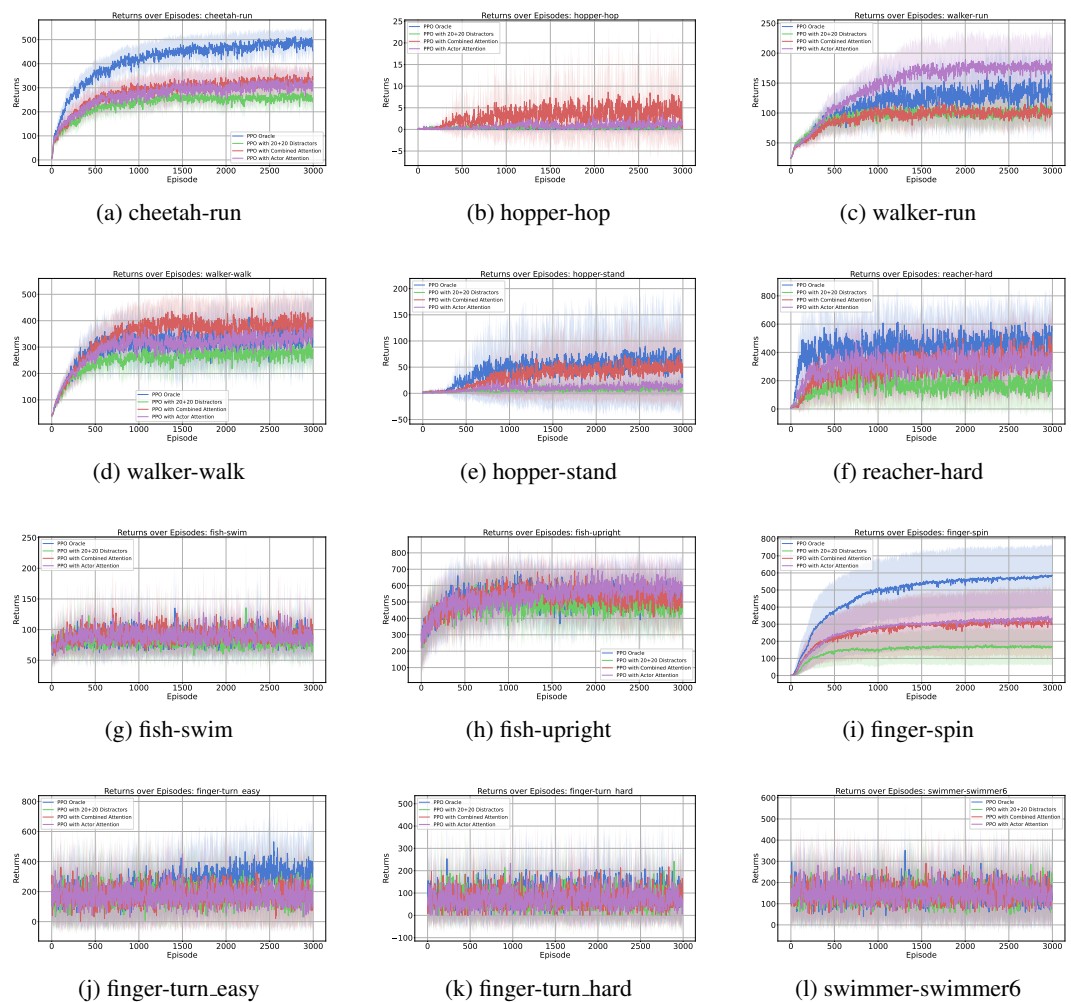

Figure 6: Performance of PPO with and without the proposed observation-conditioned gating mechanism across various DM Control Suite tasks. Plots are averaged over 10 random seeds

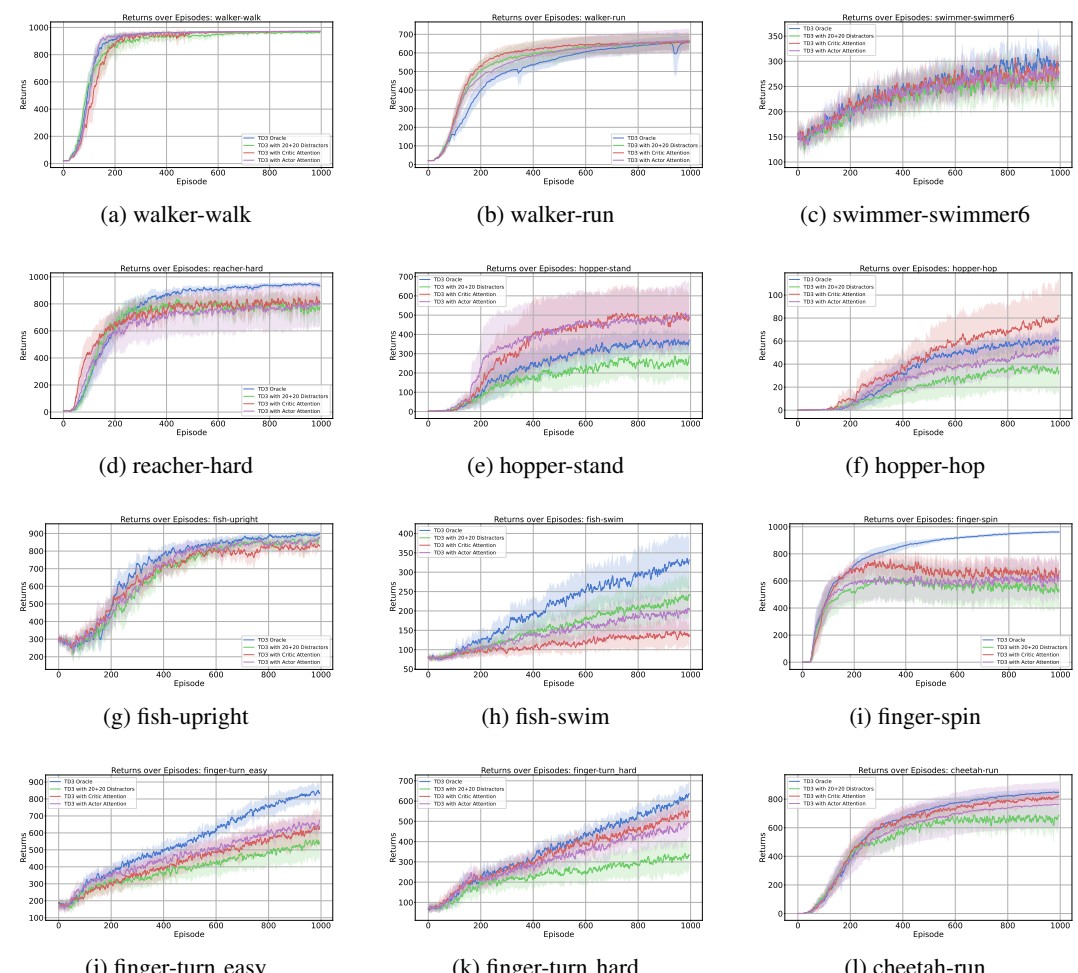

Figure 7: Performance of TD3 on the DM Control Suite tasks. Plots are averaged over 10 random seeds

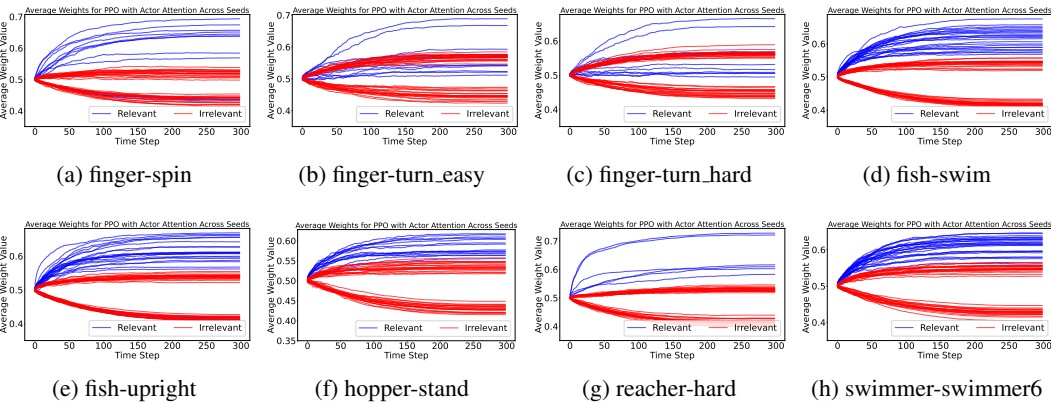

Figure 8: Masks for PPO. The task-relevant variables are plotted in blue while the task-irrelevant variables are plotted in red

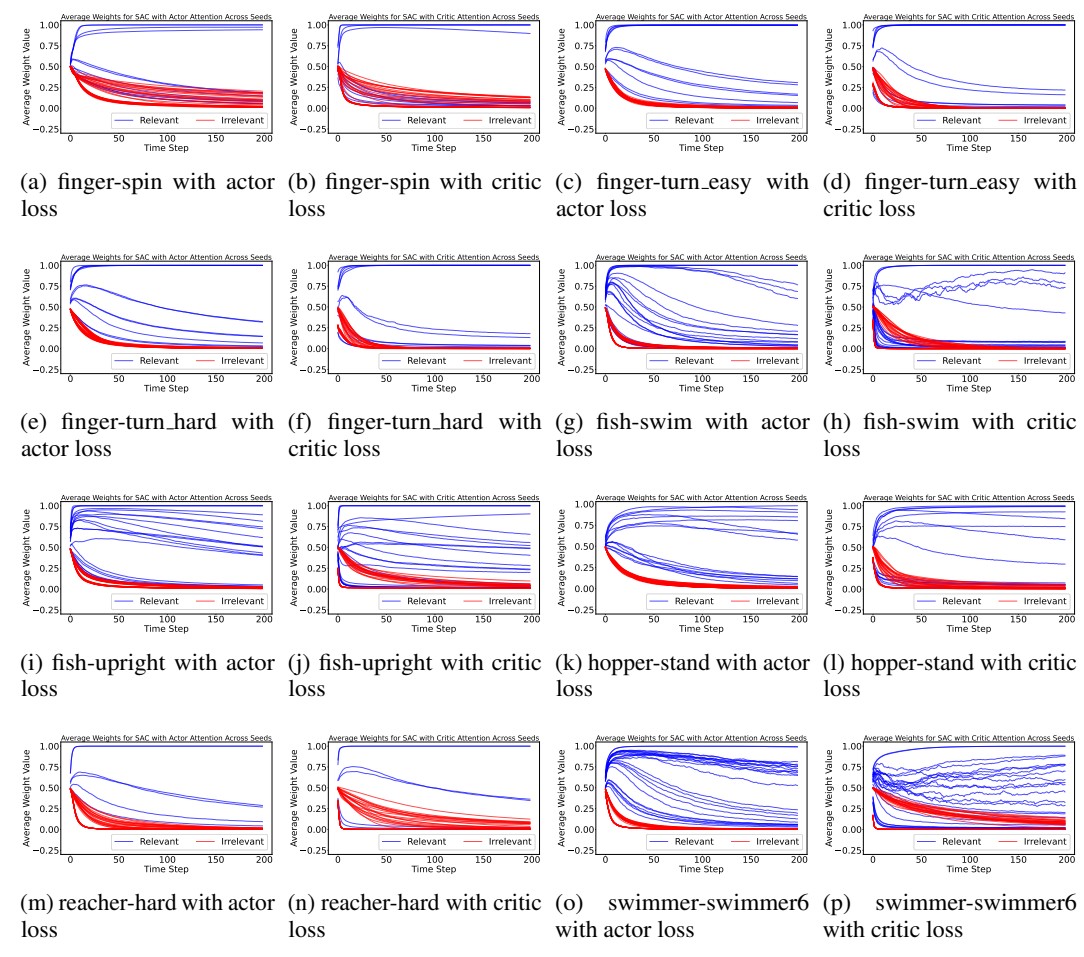

(a) finger-spin with actor loss

(b) finger-spin with critic loss

(c) finger-turn_easy with actor loss

(d) finger-turn_easy with critic loss

(e) finger-turn_hard with actor loss

(f) finger-turn_hard with critic loss

(g) fish-swim with actor loss

(h) fish-swim with critic loss

(i) fish-upright with actor loss

(j) fish-upright with critic loss

(k) hopper-stand with actor loss

(l) hopper-stand with critic loss

(m) reacher-hard with actor loss

(n) reacher-hard with critic loss

(o) swimmer-swimmer6 with actor loss

(p) swimmer-swimmer6 with critic loss

Figure 9: Masks for SAC

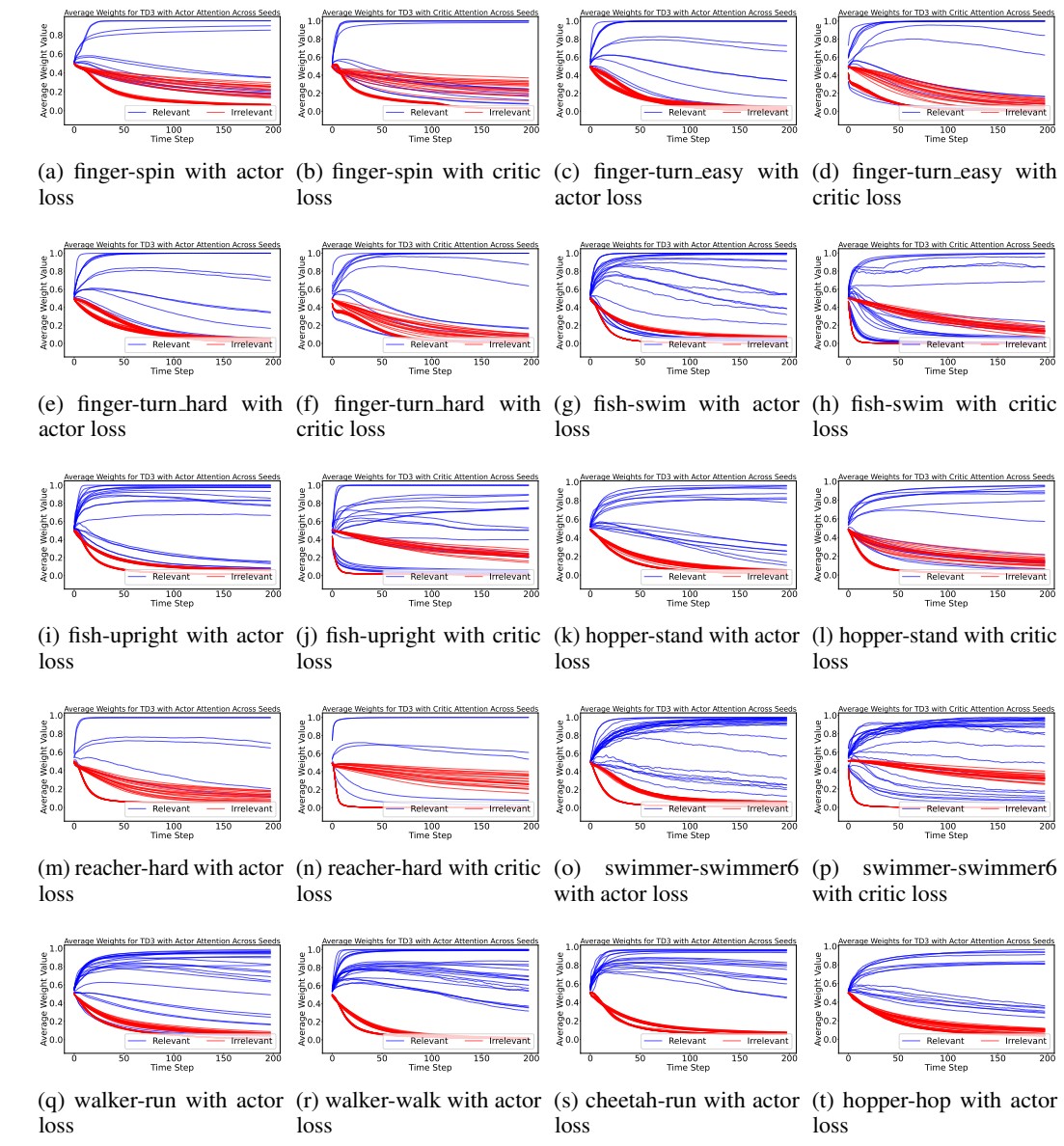

(a) finger-spin with actor loss

(b) finger-spin with critic loss

(c) finger-turn_easy with actor loss

(d) finger-turn_easy with critic loss

(e) finger-turn_hard with actor loss

(f) finger-turn_hard with critic loss

(g) fish-swim with actor loss

(h) fish-swim with critic loss

(i) fish-upright with actor loss

(j) fish-upright with critic loss

(k) hopper-stand with actor loss

(l) hopper-stand with critic loss

(m) reacher-hard with actor loss

(n) reacher-hard with critic loss

(o) swimmer-swimmer6 with actor loss

(p) swimmer-swimmer6 with critic loss

(q) walker-run with actor loss

(r) walker-walk with actor loss

(s) cheetah-run with actor loss

(t) hopper-hop with actor loss

Figure 10: Masks for TD3

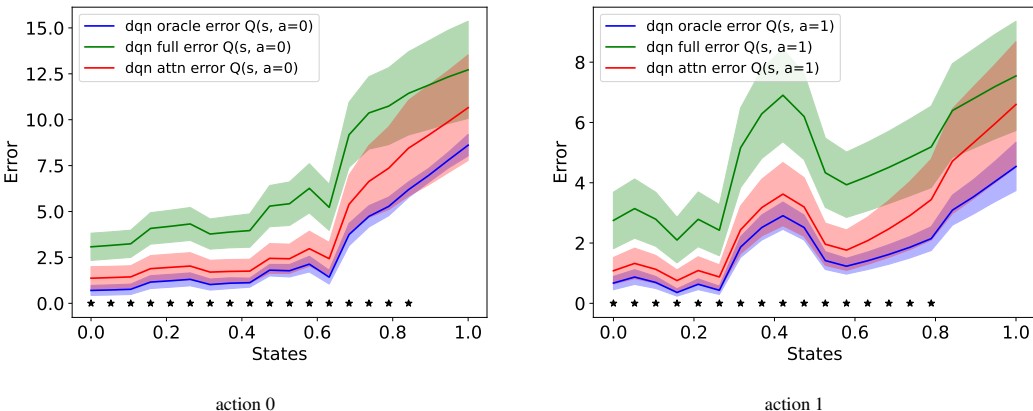

Figure 11: Mean squared error of Q-value estimation for action 0 and 1 across oracle, full, and attention-based agents with sigmoid temperature 10.

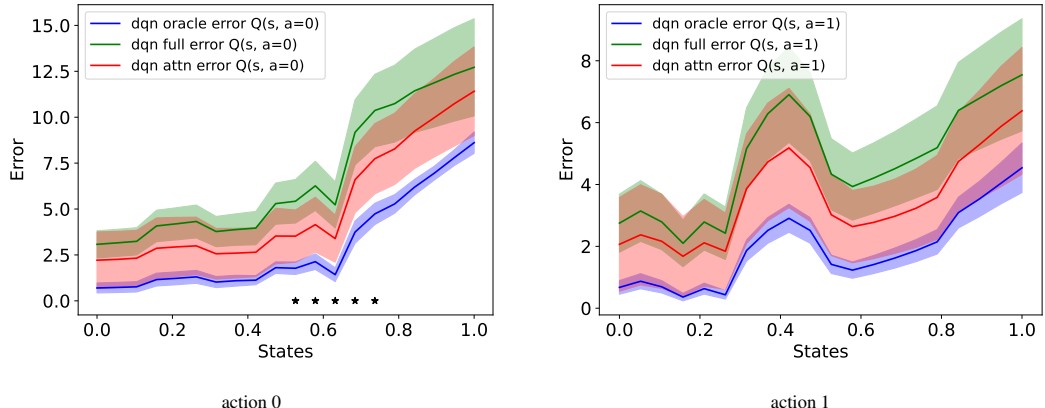

Figure 12: Mean squared error of Q-value estimation for action 0 and 1 across oracle, full, and attention-based agents with sigmoid temperature 50.

## B    GRADIENT UPDATES DERIVATION

Let the true data generating function be given by

$$y(X) = mX + c,$$

and the oracle function with learnable weights $w_0$ and $w_1$ be

$$f(x) = w_0 + w_1 x.$$

The gradient descent update rule for the weight vector $\boldsymbol{w} = \langle w_0, w_1 \rangle$ is

$$\boldsymbol{w}^{t+1} = \boldsymbol{w}^t - \eta \mathbb{E}_{X \sim P(X)} \left[ \nabla_{\boldsymbol{w}^t} (f(X) - Y)^2 \right].$$

$$\boldsymbol{w}^{t+1} = \boldsymbol{w}^t - \eta \mathbb{E}_X \left[ \nabla_{\boldsymbol{w}} (f(X) - Y)^2 \right] \tag{12}$$

$$= \boldsymbol{w}^t - 2\eta \mathbb{E}_X \left[ (f_{\boldsymbol{w}}(X) - Y) \nabla_{\boldsymbol{w}} f(X) \right] \tag{13}$$

### B.1    ORACLE UPDATES

Assuming the input $X$ follows a standard normal distribution, $X \sim \mathcal{N}(0, 1)$, we have

$$\mathbb{E}[X] = 0$$

and

$$\mathrm{Var}(X) = \mathbb{E}[X^2] - (\mathbb{E}[X])^2 \implies \mathbb{E}[X^2] = 1.$$

We denote the expectation over $X \sim \mathcal{N}(0, 1)$ as $\mathbb{E}[(\cdot)]$. The gradient descent update for the weights $\langle w_0, w_1 \rangle$ is derived as follows:

$$
\begin{aligned}
\langle w_0^{t+1}, w_1^{t+1} \rangle &= \langle w_0^t, w_1^t \rangle - \eta \mathbb{E} \left[ \nabla_{\mathbf{w}} \left( (w_0^t + w_1^t X) - (mX + c) \right)^2 \right] \\
&= \langle w_0^t, w_1^t \rangle - \eta \mathbb{E} \left[ \left( (w_0^t + w_1^t X) - (mX + c) \right) \nabla_{\mathbf{w}} (w_0^t + w_1^t X) \right] \\
&= \langle w_0^t, w_1^t \rangle - \eta \mathbb{E} \left[ \left( (w_0^t + w_1^t X) - (mX + c) \right) \left\langle \frac{\partial (w_0^t + w_1^t X)}{\partial w_0^t}, \frac{\partial (w_0^t + w_1^t X)}{\partial w_1^t} \right\rangle \right] \\
&= \langle w_0^t, w_1^t \rangle - \eta \mathbb{E} \left[ \left( (w_0^t + w_1^t X) - (mX + c) \right) \langle 1, X \rangle \right] \\
&= \langle w_0^t, w_1^t \rangle - \eta \mathbb{E} \left[ \langle w_0^t + w_1^t X - mX - c, w_0^t X + w_1^t X^2 - mX^2 - cX \rangle \right] \\
&= \langle w_0^t, w_1^t \rangle - \eta \left( \langle \mathbb{E}[w_0^t + w_1^t X - mX - c], \mathbb{E}[w_0^t X + w_1^t X^2 - mX^2 - cX] \rangle \right) \\
&= \langle w_0^t, w_1^t \rangle - \eta \left( \langle w_0^t - c, w_1^t - m \rangle \right) \\
&= \langle (1 - \eta) w_0^t + \eta c, (1 - \eta) w_1^t + \eta m \rangle.
\end{aligned}
$$

### B.2    FULL UPDATES

The model is given by

$$f(x, d) = w_0 + w_1 x + w_2 d.$$

Let the noise variable $D$ be uniformly distributed between 0 and 1, $D \sim \mathcal{U}(0, 1)$. Then, its expected value and variance are

$$\mathbb{E}[D] = \frac{1}{2}$$

and

$$\mathrm{Var}(D) = \mathbb{E}[D^2] - (\mathbb{E}[D])^2 = \frac{1}{12},$$

which implies

$$\mathbb{E}[D^2] = \frac{1}{3}.$$

Since $X \sim \mathcal{N}(0, 1)$ and $D \sim \mathcal{U}(0, 1)$ are sampled independently, their covariance is zero, and thus

$$\mathbb{E}[XD] = \mathbb{E}[X]\mathbb{E}[D] = 0 \cdot \frac{1}{2} = 0.$$

We denote the expectation over $X \sim \mathcal{N}(0,1)$ and $D \sim \mathcal{U}(0,1)$ as $\mathbb{E}[(\cdot)]$. The gradient descent update for the weight vector $\mathbf{w} = \langle w_0, w_1, w_2 \rangle$ is

$$
\begin{aligned}
\langle w_0^{t+1}, w_1^{t+1}, w_2^{t+1} \rangle &= \langle w_0^t, w_1^t, w_2^t \rangle - \eta \mathbb{E}\left[ \nabla_{\mathbf{w}^t} \left\{ (w_0^t + w_1^t X + w_2^t D) - (mX + c) \right\}^2 \right] \\
&= \langle w_0^t, w_1^t, w_2^t \rangle - \eta \mathbb{E}\left[ \left\{ (w_0^t + w_1^t X + w_2^t D) - (mX + c) \right\} \nabla_{\mathbf{w}^t}(w_0^t + w_1^t X + w_2^t D) \right] \\
&= \langle w_0^t, w_1^t, w_2^t \rangle - \eta \mathbb{E}\left[ \left\{ (w_0^t + w_1^t X + w_2^t D) - (mX + c) \right\} \left\langle \frac{\partial f}{\partial w_0^t}, \frac{\partial f}{\partial w_1^t}, \frac{\partial f}{\partial w_2^t} \right\rangle \right] \\
&= \langle w_0^t, w_1^t, w_2^t \rangle - \eta \mathbb{E}\left[ \left\{ (w_0^t + w_1^t X + w_2^t D) - (mX + c) \right\} \langle 1, X, D \rangle \right] \\
&= \langle w_0^t, w_1^t, w_2^t \rangle - \eta \mathbb{E}\left[ \langle w_0^t + w_1^t X + w_2^t D - mX - c, \right. \\
&\qquad \left. w_0^t X + w_1^t X^2 + w_2^t XD - mX^2 - cX, w_0^t D + w_1^t XD + w_2^t D^2 - mXD - cD \rangle \right] \\
&= \langle w_0^t, w_1^t, w_2^t \rangle - \eta \left\{ \langle \mathbb{E}[w_0^t + w_1^t X + w_2^t D - mX - c], \right. \\
&\qquad \left. \mathbb{E}[w_0^t X + w_1^t X^2 + w_2^t XD - mX^2 - cX], \mathbb{E}[w_0^t D + w_1^t XD + w_2^t D^2 - mXD - cD] \rangle \right\} \\
&= \langle w_0^t, w_1^t, w_2^t \rangle - \eta \left\{ \left\langle w_0^t + \frac{w_2^t}{2} - c, w_1^t - m, \frac{w_0^t}{2} + \frac{w_2^t}{3} - \frac{c}{2} \right\rangle \right\} \\
&= \left\langle (1-\eta)w_0^t - \frac{\eta}{2} w_2^t + \eta c, (1-\eta)w_1^t + \eta m, \left(1 - \frac{\eta}{3}\right) w_2^t - \frac{\eta}{2} w_0^t + \frac{\eta c}{2} \right\rangle.
\end{aligned}
$$

### B.3 ATTENTION UPDATES

The model with attention mechanisms is

$$
f(x, d) = w_0 + w_1(x\, \sigma(\phi_1)) + w_2(d\, \sigma(\phi_2)),
$$

where $\sigma(\phi) = \frac{1}{1+e^{-\phi}}$ is the sigmoid function, and its derivative is

$$
\frac{d}{d\phi}\sigma(\phi) = \sigma(\phi)(1 - \sigma(\phi)).
$$

The gradient descent update for the parameter vector $\langle w_0, w_1, w_2, \phi_1, \phi_2 \rangle$ is

$$
\begin{aligned}
\langle w_0^{t+1}, w_1^{t+1}, w_2^{t+1}, \phi_1^{t+1}, \phi_2^{t+1} \rangle &= \langle w_0^t, w_1^t, w_2^t, \phi_1^t, \phi_2^t \rangle \\
&\quad - \eta \mathbb{E}\left[ \nabla_{\boldsymbol{w}, \boldsymbol{\phi}} \left\{ (w_0 + w_1(X\, \sigma(\phi_1)) + w_2(D\, \sigma(\phi_2))) - (mX + c) \right\}^2 \right] \\
&= \langle w_0^t, w_1^t, w_2^t, \phi_1^t, \phi_2^t \rangle \\
&\quad - \eta \mathbb{E}\left[ \left\{ f(X, D) - (mX + c) \right\} \nabla_{\boldsymbol{w}, \boldsymbol{\phi}} f(X, D) \right],
\end{aligned}
$$

where the gradient of $f(X, D)$ with respect to the parameters is

$$
\nabla_{\boldsymbol{w}, \boldsymbol{\phi}} f(X, D) = \langle 1,\ X\, \sigma(\phi_1),\ D\, \sigma(\phi_2),\ w_1\, X\, \sigma(\phi_1)(1 - \sigma(\phi_1)),\ w_2\, D\, \sigma(\phi_2)(1 - \sigma(\phi_2)) \rangle.
$$

Let the error be denoted as $\text{error} = f(X, D) - (mX + c)$. Taking expectations, we derive the update rules for each parameter.

**For the $w_0$ component:**

$$
\mathbb{E}[\text{error} \cdot (1)] = w_0 + \frac{w_2\, \sigma(\phi_2)}{2} - c,
$$

leading to the update

$$
w_0^{t+1} = w_0 - \eta \left( w_0 + \frac{w_2\, \sigma(\phi_2)}{2} - c \right).
$$

**For the $w_1$ component:**

$$
\mathbb{E}[\text{error} \cdot (X\, \sigma(\phi_1))] = \sigma(\phi_1)(w_1\, \sigma(\phi_1) - m),
$$

leading to the update

$$
w_1^{t+1} = w_1 - \eta \left( w_1\, \sigma(\phi_1)^2 - m\, \sigma(\phi_1) \right).
$$

**For the $w_2$ component:**

$$
\mathbb{E}[\text{error} \cdot (D\, \sigma(\phi_2))] = \sigma(\phi_2) \left( \frac{w_0}{2} + \frac{w_2\, \sigma(\phi_2)}{3} - \frac{c}{2} \right),
$$

leading to the update

$$w_2^{t+1} = w_2 - \eta\,\sigma(\phi_2)\left(\frac{w_0}{2} + \frac{w_2\,\sigma(\phi_2)}{3} - \frac{c}{2}\right).$$

**For the $\phi_1$ component:**

$$\mathbb{E}[\text{error} \cdot (w_1\,X\,\sigma(\phi_1)(1 - \sigma(\phi_1)))] = w_1\,\sigma(\phi_1)(1 - \sigma(\phi_1))(w_1\,\sigma(\phi_1) - m),$$

leading to the update

$$\phi_1^{t+1} = \phi_1 - \eta\,w_1\,\sigma(\phi_1)(1 - \sigma(\phi_1))(w_1\,\sigma(\phi_1) - m).$$

**For the $\phi_2$ component:**

$$\mathbb{E}[\text{error} \cdot (w_2\,D\,\sigma(\phi_2)(1 - \sigma(\phi_2)))] = w_2\,\sigma(\phi_2)(1 - \sigma(\phi_2))\left(\frac{w_0}{2} + \frac{w_2\,\sigma(\phi_2)}{3} - \frac{c}{2}\right),$$

leading to the update

$$\phi_2^{t+1} = \phi_2 - \eta\,w_2\,\sigma(\phi_2)(1 - \sigma(\phi_2))\left(\frac{w_0}{2} + \frac{w_2\,\sigma(\phi_2)}{3} - \frac{c}{2}\right).$$

In summary, the gradient descent update rules for the attention model are:

$$w_0^{t+1} = w_0 - \eta\left(w_0 + \frac{w_2\,\sigma(\phi_2)}{2} - c\right),$$

$$w_1^{t+1} = w_1 - \eta\left(w_1\,\sigma(\phi_1)^2 - m\,\sigma(\phi_1)\right),$$

$$w_2^{t+1} = w_2 - \eta\,\sigma(\phi_2)\left(\frac{w_0}{2} + \frac{w_2\,\sigma(\phi_2)}{3} - \frac{c}{2}\right),$$

$$\phi_1^{t+1} = \phi_1 - \eta\,w_1\,\sigma(\phi_1)(1 - \sigma(\phi_1))(w_1\,\sigma(\phi_1) - m),$$

$$\phi_2^{t+1} = \phi_2 - \eta\,w_2\,\sigma(\phi_2)(1 - \sigma(\phi_2))\left(\frac{w_0}{2} + \frac{w_2\,\sigma(\phi_2)}{3} - \frac{c}{2}\right).$$

## C  HYPERPARAMETERS

### C.1  FOR SAC

Table 1: Key hyperparameters and architecture details of our SAC implementation.

| Component | Setting |
|---|---|
| **Environment** | |
| Environment | DM Control Suite |
| Observation Dim | Original + 20 controllable + 20 uncontrollable distractors |
| Action Space | Continuous (Box) |
| **Actor Network** | |
| Architecture | MLP: 256-256, ReLU |
| **Critic Networks (Q1/Q2)** | |
| Architecture | MLP: 256-256, ReLU |
| Target Update | Soft update with $\tau = 0.005$ |
| **Training Setup** | |
| Replay Buffer Size | $5 \times 10^6$ |
| Batch Size | 256 |
| Learning Starts | 10,000 steps |
| Total Steps | 1M |
| Policy LR | $3 \times 10^{-4}$ |
| Critic LR | $1 \times 10^{-3}$ |
| Optimizer | Adam |
| Max Grad Norm | 10 |
| **Entropy Tuning** | |
| $\alpha$ Tuning | Enabled (learned) |
| **Distractors** | |
| Controllable | Linear in action + bias |
| Uncontrollable | Random + bias |
| **Evaluation & Logging** | |
| Eval Episodes | 50 (continuous) |

## C.2 PPO

Table 2: Key hyperparameters and architecture details of our PPO implementation.

| Component | Setting |
|---|---|
| **Environment** | |
| Environment | DM Control Suite |
| Observation Dim | Original + 20 controllable + 20 uncontrollable distractors |
| Action Space | Continuous (Box) |
| **Agent Architecture** | |
| Policy/Value Network | Shared MLP: 64-64, `Tanh` activations |
| **Training Setup** | |
| Total Timesteps | 3M |
| Rollout Length | 2000 steps |
| Mini-batches | 40 |
| Update Epochs | 10 |
| Optimizer | Adam, LR $= 3 \times 10^{-4}$ |
| Annealed LR | Yes |
| Gradient Clipping | Max norm $= 0.5$ |
| **PPO Settings** | |
| GAE Lambda | 0.95 |
| Discount Factor $\gamma$ | 0.99 |
| Advantage Normalization | Enabled |
| Clip Coefficient | 0.2 |
| Clip Value Loss | Enabled |
| Entropy Coefficient | 0.0 |
| Value Loss Coef | 0.5 |
| Target KL | None |
| **Distractor Variables** | |
| Controllable | Linear in action + fixed bias |
| Uncontrollable | Random noise + fixed bias |
| **Evaluation & Logging** | |
| Eval | Returns from rollouts |

## C.3 TD3

Table 3: Key hyperparameters and architectural details of our TD3 implementation with shared attention.

| Component | Setting |
|---|---|
| **Environment** | |
| Environment | DM Control Suite |
| Observation Dim | Original + 20 controllable + 20 uncontrollable distractors |
| Action Space | Continuous (Box) |
| **Actor Network** | |
| Architecture | MLP: 256-256, ReLU |
| **Critic Networks (Q1, Q2)** | |
| Architecture | MLP: 256-256, ReLU |
| Target Networks | Soft update with $\tau = 0.005$ |
| **Training Setup** | |
| Total Timesteps | 1M |
| Learning Starts | 25K steps |
| Batch Size | 256 |
| Replay Buffer | Size = $10^6$ transitions |
| Optimizer | Adam, LR = $3 \times 10^{-4}$ |
| Gradient Clipping | Not used |
| **TD3-Specific Settings** | |
| Policy Delay | 2 steps |
| Target Policy Noise | Std = 0.2, Clipped at 0.5 |
| Exploration Noise | Std = 0.1 (added to actor output) |
| **Distractor Variables** | |
| Controllable | Linear in action + fixed bias |
| Uncontrollable | Random noise + fixed bias |
| **Evaluation & Logging** | |
| Eval Episodes | 50 continuous episodes per checkpoint |

## D  CODE

Anonymous GitHub link:

```
https://anonymous.4open.science/r/submission-41DF
```

