# OpenReview forum: "Architectural Inductive Biases Can Be Enough for State Abstraction in Deep Reinforcement Learning"
_ICLR.cc/2026/Conference — Submitted to ICLR 2026_

### Official Review · Reviewer_JjLf · 2025-10-26

**Soundness:** 4
**Presentation:** 2
**Contribution:** 2
**Rating:** 4
**Confidence:** 4

**Summary:**

The paper proposes an architectural modification for neural-network based policies that improves reinforcement learning performance in the presence of distractors. The problem setting is a factored state MDP, where parts of the state space do not contribute to the reward and transition dynamics. The authors propose learning an attention mask purely from the reward through either the actor or critic loss. Experiments on the modified DeepMind Control Suite show that the proposed masking often improves performance over vanilla algorithms (SAC, TD3, PPO). While the paper is well structured and provides ample empirical evidence, some concerns remain regarding the motivation and evaluation.

**Strengths:**

- The proposed mask is simple, easy to integrate with existing algorithms, and does not involve computational overhead or additional training steps.
- The "Controlled Analysis" of the masking approach in supervised learning and toy MDPs provides some intuition on how distractors impede learning and why the masking may mitigate this.
- The evaluation is conducted using many different tasks from the DeepMind Control Suite as well as several state-of-the-art actor critic RL algorithms. Statistical relevance of the results is shown by using several random seeds for each task and algorithm.
- Both structure and writing in the paper are very good.

**Weaknesses:**

- The proposed architecture seems to be mainly applicable to settings with factored state spaces, meaning the observation is a vector. Most existing work in the area of state abstraction focuses on visual domains, which seem to be more relevant. Given a state space with a time-invariant factorization, it seems more promising to identify the irrelevant states - e.g., using SHAP values - and then train agents using a reduced observation space. Not only is the proposed approach not tested on visual domains (e.g., Distracting Control Suite), it also does not seem applicable to such tasks. The motivation of the paper should be improved w.r.t. to these points.
- The authors compare their approach to only one alternative - MaDi - which was devised for visual domains. In the original MaDi paper, the masking network consists of a small convolutional neural network to process RGB images. If a similar architecture was used in this work, this would be ill-suited for simple vector observations. Therefore, we suggest comparing to other approaches designed for factorized state spaces like the CBM proposed in (Wang et al. 2024).
- The presentation could be improved by increasing the size of Fig. 1, 3, and 4.

**Questions:**

- Why is the factored MDP formulated for discrete states and actions? The tasks in the experiments seem to feature continuous state and action spaces.
- Why is PPO not paired with MaDi in the evaluation?

---

> ### Author Response · Authors · 2025-11-27
>
> Thank you for a detailed feedback. We address your concerns below.
>
> **On factored state spaces.**
> Our choice of factored state spaces for the main results is motivated by a variety of observations. First, this setting is consistent with other recent works such as [1, 2] that aim to learn abstractions but do not consider pixel-based environments. This is partly because it allows us to design a quantifiable “ground-truth” for abstraction tasks and isolates the contribution of the proposed architecture to distractor suppression and improved performance on control tasks. From an empirical point of view, these settings are not a simplification in any form and remain non-trivial, even for the most widely used algorithms. Furthermore, a broad range of real-world RL applications still operate over spaces where the state is not an image but a collection of sensor readings, system variables, or engineered features from high-dimensional raw data. For example, in robotics (joint angles, torques, velocities, …), autonomous driving (object tracks, distances, relative velocities, …), finance (prices, technical indicators, …), healthcare (vitals, demographics, lab measurements, …), operations research (queue lengths, service times, resource availability), and so on. These domains often suffer from exactly the problem we study: an abundance of spurious, redundant, or weakly correlated features that degrade learning in finite data regimes.
>
> With respect to SHAP-like approaches, there are a few important limitations. First, SHAP and similar methods are post-hoc analysis tools. They do not directly influence the learning dynamics of the agent during training. Second, SHAP is usually defined for static input–output mappings, whereas in policy gradient methods, the data distribution depends on the current policy, which itself changes at every update. As a result, any score computed at time t may not remain valid even a few updates later. Third, SHAP assumes access to a reasonably well-trained and stable model. In the presence of distractors, the core problem is precisely that the model fails to train effectively in the first place. A method that depends on a learned model to determine relevance therefore presupposes the solution to the very problem we are attempting to address. Fourth, integrating SHAP into the training loop is computationally prohibitive for deep RL. Even in supervised settings, SHAP could be expensive due to its combinatorial nature and sampling approximations. In RL, this cost would need to be paid repeatedly at every policy update step.
> More fundamentally, our work is not about feature selection as a preprocessing step, it is about inducing abstraction during learning. This is a crucial difference. Our mask is not fitted using a supervised relevance signal, labels, or explicit notion of importance. It emerges purely from optimization pressure induced by long-term return, which is precisely what makes it relevant for RL.
> We will strengthen the motivation to make this explicit: our contribution is not positioned as a competitor to context-adaptive visual abstraction methods, but as a foundational building block that demonstrates, in a clean and controlled manner, that reward-driven abstraction is viable.
>
> **On MaDi.**
> The CNN used in MaDi was replaced by an MLP in our experiments to adapt to vector observations for a fair comparision. A direct comparision with prior methods such as [1, 2] is not meaningful, as those approaches are model-based and explicitly rely on learned dynamics or causal structure while our method is entirely model-free. We did not benchmark MaDi against PPO because the original work proposing MaDi reports results exclusively with off-policy methods. We suspect this choice is deliberate, as the introduction of augmented distractors makes representation learning significantly more challenging, requiring the agent to simultaneously learn an effective gating mechanism alongside the control policy. In this setting, on-policy methods like PPO, which are inherently more data-inefficient, are at a disadvantage compared to off-policy methods that reuse experience to train both the policy and the masking more efficiently.
>
> We note the suggestion about figure sizes and will make appropriate changes.
>
> [1] Wang, Zizhao, et al. "Causal dynamics learning for task-independent state abstraction." arXiv preprint arXiv:2206.13452 (2022).
>
> [2] Wang, Zizhao, et al. "Building minimal and reusable causal state abstractions for reinforcement learning." Proceedings of the AAAI Conference on Artificial Intelligence. Vol. 38. No. 14. 2024.

---

### Official Review · Reviewer_RBxV · 2025-10-29

**Soundness:** 3
**Presentation:** 3
**Contribution:** 2
**Rating:** 4
**Confidence:** 3

**Summary:**

This paper explores state abstraction in deep reinforcement learning. The focus of the work is to better understand what minimal ingredients might be needed to facilitate learning a state abstraction; many prior approaches rely on additional losses or auxiliary tasks, but it is not clear these are strictly necessary to learn a state abstraction. To understand this question, the work proposes a simple soft gating mechanism to the input features, captured by a per-feature gating mechanism, $\sigma$. In this way, if $\sigma$ maps a feature to zero, this encodes the fact that the learner should treat this feature as a distractor. Notably, $\sigma$ is modelled as a soft-gating mechanism rather than a strict one, which facilitates smooth updates to $\sigma$ based on the standard RL loss. The primary experiments come in two forms. First, variants of SAC are contrasted that make use of this learned gating mechanism on just the actor, just the critic, both, an and an oracle (among others). Normalized returns are compared across these methods. Then, in Figure 2, the weight values for the gating mechanism are visualized on a per-feature level and grouped according to whether the feature is a distractor or not. Then, two additional analyses are carried out: the first is in a linear regression setting with distractors (Props 1-3), and the second is a policy evaluation experiment with distractors (Fig. 3 and 4).

**Strengths:**

The paper has several strengths:
1. The work sets out to study a compelling question: are the standard RL loss and architectures sufficient on their own to learn a state abstraction? While there are some aspects of this question that could be framed in a slightly more precise way, I find the question to be valuable in its own right.
2. The approach is relatively simple: a soft gating mechanism as described by $\sigma$ is lightweight and can be incorporated into many kinds of deep RL architectures and learning settings.
3. The experimental design is clean and affords quick contact with intuition. I especially found Figure 2 to be a compelling experiment in light of the question at the heart of the paper.

**Weaknesses:**

There are two primary weaknesses (PW.x) that I believe should be discussed further:
- PW.1: The central question is slightly imprecise, and therefore leaves some ambiguity as to the aim and takeaways of the work
- PW.2: The gating mechanism relies on a feature space that is split between either relevant or irrelevant features.

In more detail:

W.1: I believe more could be done to clarify the central question of the work. My suggestion is to dig into what, specifically, it would mean for the question to have a negative answer, and why something like DQN is not already evidence of a positive answer.

In more detail: In some sense, the question being posed has an obvious answer, as evidenced by the advent of deep RL: DQN, Rainbow, and most early deep RL methods were among the first RL methods to achieve high performance in environments with rich input spaces such as images. The premise, in a way, of deep RL was that state abstraction and/or representation learning can be carried out implicitly in the service of value prediction (or control, or other broader goals of the learner). Of course as the present paper points out, much research has been dedicated to trying to learn good representations explicitly through the use of extraneous components like new losses, extra networks (auto-encoders), auxiliary tasks, and so on.

At present I do not believe it is clear what separates the core question ("Are architectural choices and the RL objective alone sufficient to learn abstract state representations?") from what DQN already achieved early on. I believe there is a reasonable answer to this, but I also believe that answer is not quite present in the current paper.

A related point, but one standard definition of state abstraction is in terms of state aggregation (as in Li, Walsh, Littman, 2006). Aggregation is a simple form of function approximation. Since neural networks are much more general forms of function approximators, it should be clear they are capable of state aggregation. So, I am again wondering: what is the precise hypothesis, and in what way does something like DQN not already provide evidence for that hypothesis?

W.2: The function $\sigma$ is relatively simple, and is perhaps only salient in settings when the input features are clearly strictly a distractor, or not, for all time. This seems less applicable in a domain like Atari, where input features are pixels, and pixel values can either take on relevance or not depending on the context. How do you see this framing extending beyond the case when explicit distractors are added, to something more like the Atari setup?


A short note: The proposed gating mechanism is closely related to soft state aggregation (Sing, Jaakkola, Jordan, 1994). I believe some discussion of this connection would be useful.


References:
- Singh, S., Jaakkola, T., & Jordan, M. (1994). Reinforcement learning with soft state aggregation. Advances in neural information processing systems, 7.

**Questions:**

My primary questions stem from the two major questions of the work.

Q1. What specifically does it mean for a deep RL method to have learned a state abstraction?

I can imagine a few responses:
- (i) It has learned to ignore explicit distractors that are included in the input space.
- (ii) It has learned any compression of the state space
- (iii) It has learned a useful compression of the state space, in line with some of the state abstraction families from Li, Walsh, Littman (2006).
- (iv) It has learned a meaningful state representation that facilitates effective decision making in the environment in question.

I believe this work is focusing on (i). But if so, it should make that clear, and illustrate that at least some past method does not already do this.

Q2: How reliant is the study on the existence of explicit distractors? In other words, how would the framing apply in settings like Atari, where input features are just pixels, and their relevance can evolve over time?

---

> ### Author Response · Authors · 2025-11-27
>
> We thank the reviewer for the insightful comments, which touch on some of the fundamental questions underlying this work. We address them as follows:
>
>
> **On learning state abstractions in deep RL.**
> To have learned a state abstraction ideally would mean that the agent has identified the most compact representation of the state space that still lets it learn the optimal policy.
>
> The question about whether deep RL already does this is partly answered by our work and has been partly answered in prior work such as [3]. They show that deep neural nets indeed do perform some compression of the input space with variables irrelevant for prediction being suppressed to an extent during training. However, environments such as Atari or MuJoCo do not generally contain excessive information that is unnecessary for the agent to accomplish the current task, something a real-world robotic agent might face. And hence, the capacity of neural architectures to deal with a state space where a large subspace is uninformative, has not been challenged in these domains which highlights a significant gap that motivates our study of such settings. We show that under the additional augmented state space, the standard architectures struggle to simultaneously find an optimal abstraction and a policy for a variety of deep RL algorithms. This implies that these algorithms have not been able to extract the original set of features with the given architectures because if they had, they would have learnt the optimal policy similar to the oracle as the hardness of policy learning is not the bottleneck to performance. They fail to learn a good policy precisely because of inefficient state abstraction. We will make this discussion more clear in the text.
>
>
> **On factored state spaces.**
> Our choice of factored state spaces for the main results is motivated by a variety of observations. First, this setting is consistent with other recent works such as [1, 2] that aim to learn abstractions but do not consider pixel-based environments. This is partly because it allows us to design a quantifiable “ground-truth” for abstraction tasks and isolates the contribution of the proposed architecture to distractor suppression and improved performance on control tasks. From an empirical point of view, these settings are not a simplification in any form and remain non-trivial, even for the most widely used algorithms. Furthermore, a broad range of real-world RL applications still operate over spaces where the state is not an image but a collection of sensor readings, system variables, or engineered features from high-dimensional raw data. For example, in robotics (joint angles, torques, velocities, …), autonomous driving (object tracks, distances, relative velocities, …), finance (prices, technical indicators, …), healthcare (vitals, demographics, lab measurements, …), operations research (queue lengths, service times, resource availability), and so on. These domains often suffer from exactly the problem we study: an abundance of spurious, redundant, or weakly correlated features that degrade learning in finite data regimes.
>
> **Extension to visual domains.**
> While the study is reliant on explicit distractors for its core results, it is more generally about the concept that in many decision-making environments, only a subset of the available state information is causally relevant for optimal control at any given time.
>
> The proposed method could potentially be extended to visual domains as well. One way would be to observe that an agent would like to preserve the relevant features required to solve the task in any given environment. Hence, we can apply a learned gating to the latent features produced by a convolutional encoder at the level of channels and train this gate jointly with the policy and value networks using only the RL objective. This is because each channel acts like a feature detector (edges, objects, motion, etc.). The mask then plays the role of a learned visual attention mechanism that suppresses features associated with distractors (e.g., background textures, color blobs or irrelevant objects depending on the kind of distractors) and preserves those that are predictive of long-term return. Having said that, a lot of important RL applications are still feature-based where it is not clear what the optimal set of variables are.
>
> We acknowledge the note about soft state aggregation and will include a brief discussion about the connection.
>
> [1] Wang, Zizhao, et al. "Causal dynamics learning for task-independent state abstraction." arXiv preprint arXiv:2206.13452 (2022).
>
> [2] Wang, Zizhao, et al. "Building minimal and reusable causal state abstractions for reinforcement learning." Proceedings of the AAAI Conference on Artificial Intelligence. Vol. 38. No. 14. 2024.
>
> [3] Saxe, Andrew M., et al. "On the information bottleneck theory of deep learning." Journal of Statistical Mechanics: Theory and Experiment 2019.12 (2019): 124020.

---

### Official Review · Reviewer_q333 · 2025-11-06

**Soundness:** 3
**Presentation:** 3
**Contribution:** 2
**Rating:** 4
**Confidence:** 3

**Summary:**

The authors propose methods that learn how to ignore irrelevant environmental variables without the use of auxiliary objectives. Instead, they propose an architectural change: “a learnable, observation-independent attention mask applied to the inputs of the policy and value networks and trained end-to-end using only the RL objective.” They implement and evaluate this approach in several simple environments.

**Strengths:**

The key idea (learning an attention mask) is simple and intuitive.

The authors provide a fairly extensive and compact survey of much of the important literature.

**Weaknesses:**

**Questionable assumptions and evaluation:** Practical and general methods for ignoring distractors would seem to necessarily be context-specific. That is, such methods would identify which features are (and are not) relevant to action selection in the current state, even when those features are different in different contexts. However, the authors appear to assume that relevant features are not context-specific (“…existing methods are typically context-dependent, suppressing features only locally when it negatively affects performance, whereas we use a simpler observation-independent mask.”). The fact that context-general suppression of features is successful in the authors’ experiments may have more to do with experimental methodology than with realism. That is, the authors evaluate their proposed methods by inserting features that are always irrelevant, rather than only irrelevant in some contexts.

**Artificial and limited experiments**: Section 6.1 describes a simple regression task and Section 6.2 describes experiments in which “The base environment is a custom MDP with continuous one-dimensional state space and two discrete actions.” These experimental setups seem artificial, even by the standards of traditional RL papers. The paper would be substantially improved by experiments with a greater range of more realistic problems.

**Questions:**

(none)

---

> ### Author Response · Authors · 2025-11-27
>
> Thank you for the feedback. We address your concerns as follows.
>
> **On factored state spaces.**
> Our choice of factored state spaces for the main results is motivated by a variety of observations. First, this setting is consistent with other recent works such as [1, 2] that aim to learn abstractions but do not consider pixel-based environments. This is partly because it allows us to design a quantifiable “ground-truth” for abstraction tasks and isolates the contribution of the proposed architecture to distractor suppression and improved performance on control tasks. From an empirical point of view, these settings are not a simplification in any form and remain non-trivial, even for the most widely used algorithms. Furthermore, a broad range of real-world RL applications still operate over spaces where the state is not an image but a collection of sensor readings, system variables, or engineered features from high-dimensional raw data. For example, in robotics (joint angles, torques, velocities, …), autonomous driving (object tracks, distances, relative velocities, …), finance (prices, technical indicators, …), healthcare (vitals, demographics, lab measurements, …), operations research (queue lengths, service times, resource availability), and so on. These domains often suffer from exactly the problem we study: an abundance of spurious, redundant, or weakly correlated features that degrade learning in finite data regimes.
>
> **On context-general vs. context-specificity.**
> Our discussion of context-specificity can be clarified and we will do so in the paper. By context-dependent we meant something different than observation-dependent. We are interested in identifying task-relevant features and this set of features will be constant across observations but may change as a function of the task the agent is engaged in. Since we only consider single-task settings (i.e., the standard RL problem), it is sufficient to learn a single set of task-relevant features. Our method is “context dependent” in the sense that the learnt mask is task-specific. In essence, we do agree that “Practical and general methods for ignoring distractors would seem to necessarily be context-specific”.
>
>
> Furthermore, while it is valid that image-based environments would need observation-dependent masking and MaDi targets this, our method’s superior performance in structured distractor settings is not obvious a priori and is a key novel finding of our study. If anything, one might reasonably expect the opposite. A masking system designed to handle more general distractors could be expected to perform well in the “easier” distraction setting. Our results demonstrate that for factored distractors and dense rewards, simpler architectural inductive bias with a few learnable parameters suffices, avoiding the complexity of auxiliary losses or observation-dependent modules and providing us with a simple baseline for future work on auxiliary losses or state abstractions to compare against.
>
> In summary, the work is not intended as a universal solution for all abstraction problems but as a foundational study showing that purely reward-driven masking can induce meaningful abstraction in non-trivial distractor settings and close most of the gap to oracle agents. We hope the reviewer will consider the value of this foundational step toward more general solutions.
>
>
> **Regression and MDP experiments.**
> The toy MDP (adapted from [3]) serves as a controlled environment which highlights that even in extremely simple MDPs, the error in q-value estimation by a fully connected DQN is statistically significantly different and worse than our method. This also allows us to compare different baselines against the true q-values. Similarly, linear regression with infinite-data gradient analysis allows us to isolate the mechanism behind why distractors harm optimization and why our method works. It also highlights that this effect does not arise due to a lack of data but is inherent to gradient descent, and hence also catastrophic for deep networks in RL unless mitigated by the introduction of appropriate architectural biases. So, this section is not intended to provide evidence of real-world applicability of our method. It is a theoretical and mechanistic analysis section. Our main result on DM Control tasks (which are extensively used in RL research to benchmark new methods including [3], published ICLR 2022) is in line with similar prior work in the literature.
>
>
> [1] Wang, Zizhao, et al. "Causal dynamics learning for task-independent state abstraction." arXiv preprint arXiv:2206.13452 (2022).
>
> [2] Wang, Zizhao, et al. "Building minimal and reusable causal state abstractions for reinforcement learning." Proceedings of the AAAI Conference on Artificial Intelligence. Vol. 38. No. 14. 2024.
>
> [3] Yang, Ge, Anurag Ajay, and Pulkit Agrawal. "Overcoming the spectral bias of neural value approximation." arXiv preprint arXiv:2206.04672 (2022).

---

### Official Review · Reviewer_QYMo · 2025-11-06

**Soundness:** 2
**Presentation:** 2
**Contribution:** 2
**Rating:** 4
**Confidence:** 3

**Summary:**

The authors show that architectural choices and RL objective alone are sufficient to learn abstract state representations. Specifically, they mask the inputs to the policy and value networks.

**Strengths:**

The proposed architectural improvements are simple and can be easily used together with existing RL algorithms. The authors provide theoretical insights on why an attention mask can lead to meaningful state abstractions.

**Weaknesses:**

The experiments can be strengthened by showing the proposed method scales with the the dimension of the controllable and uncontrollable distractors.

**Questions:**

1. the proposed method can be viewed as a form of regularization. how does it compare to having a dropout layer in the  actor and critic network?
2. how does varying the dimension of controllable and uncontrollable distractors affect the performance of the proposed method?
3. what is the choice of delta (in equation 4) in the experiments? could the proposed method handle larger values of delta?

---

> ### Author Response · Authors · 2025-11-27
>
> Thank you for your review and for raising important questions regarding the experimental section. We clarify these below.
>
> **Drawing parallels between our method and dropout.** We agree that there is a resemblance between our approach and dropout, and this is an insightful observation. By randomly suppressing features during training, dropout might indeed encourage the network to become less reliant on unstable or spurious inputs, and may therefore indirectly help in suppressing distractor variables over time. However, the gradients are already noisy due to bootstrapping and non-stationarity. Adding stochastic feature suppression injects additional noise into both the actor and critic. There is evidence that applying dropout naively destabilizes RL training [1].
>
> **Varying the dimension of variables.** We appreciate this suggestion, and we agree that varying the dimensionality of controllable and uncontrollable distractors is a meaningful axis for analysis. While we did not explicitly sweep over the absolute number of distractor variables, we note that performance in such settings would likely be influenced not only by the masking mechanism itself, but also by factors such as network capacity as the input dimension grows. For this reason, we believe that a more informative ablation would focus on varying the ratio of task-relevant to task-irrelevant variables, which directly captures the degree of distraction in the environment independent of input dimensionality. In our MuJoCo experiments, the native observation spaces already induce a natural range of such ratios, with distractors ranging from approximately 2x to up to 7x the dimensionality of the task-relevant state, providing a good coverage of the degree of distraction without an explicit sweep.
>
> **Handling larger delta.** Delta is 1 for our experiments. Our method does not rely on delta being small and should be able to handle a higher delta. A higher delta means higher variability of the distractor variables which will make it even easier for distractors to be detected and suppressed.
>
> [1] Hausknecht, Matthew, and Nolan Wagener. "Consistent dropout for policy gradient reinforcement learning." arXiv preprint arXiv:2202.11818 (2022).

---

### Meta-Review · Area_Chair_Lwad · 2026-01-07

**Summary:**

This paper argues that architectural inductive biases can yield meaningful state-abstractions for RL, thereby avoiding the need to develop auxiliary objectives for better abstractions/representations. They do so by proposing a learnable attention mask on the inputs to the RL network.

This is an interesting idea, but I ultimately feel this work is not yet ready for publication. There were a number of very valid concerns raised by the reviewers that were not properly addressed. There were also a number of sensible experiments suggested by the reviewers, which could have been reasonably run during the rebuttal phase. The authors instead argued (unconvincingly) why those suggested experiments were not meaningful, rather than running some of them.

One of the main issue raised by a number of reviewers was the assumption of context-independent irrelevant features. I do not feel the authors addressed this properly, mostly arguing that there are lots of environments where this property holds (even though they did not evaluate on any of these). As such, this unresolved concern indicates that either the paper is improperly scoped (i.e. they should limit their contributions to environments where context-independent irrelevant features may exist), or propose different architectural biases that can deal with context-dependent irrelevant features, so as to support the paper's main thesis more convicingly.

**Reviewer Concerns:**

I highlight the main concerns raised by reviewers.

## QYMo
- W1 (The experiments can be strengthened by showing the proposed method scales with the the dimension of the controllable and uncontrollable distractors). Not addressed
- Q1 (how does it compare to having a dropout layer in the actor and critic network?). The authors simply argued why this experiment isn't meaningful, rather than running the comparison, which is a reasonable suggestion.
- Q2 (how does varying the dimension of controllable and uncontrollable distractors affect the performance of the proposed method?) The authors simply argued why this experiment isn't meaningful, rather than running the comparison, which is a reasonable suggestion.

## q333
- W1 (Questionable assumptions and evaluation... the authors evaluate their proposed methods by inserting features that are always irrelevant, rather than only irrelevant in some contexts). The authors mainly argue that "a broad range of real-world RL applications still operate over spaces where the state is not an image", and provide some hypothetical examples where this would be the case. This is a somewhat weak response, as ideally the authors would have run evaluations on said environments, as opposed to only on artificially-altered MuJoCo experiments and toy MDPs. This is also discussed with reviewer RBxV.
- W2 (Artificial and limited experiments). Same as above.

## RBxV
- W1 (The central question is slightly imprecise). The reviewer elaborates this point quite well and in a lot of detail, in particular raising the point that "why something like DQN is not already evidence of a positive answer.". The authors respond that "environments such as Atari or MuJoCo do not generally contain excessive information that is unnecessary for the agent to accomplish the current task, something a real-world robotic agent might face." This is not true (think of the game Pong, where most of the pixels are irrelevant for control but, importantly, _which_ pixels are irrelevant depend on the current frame). It is interesting that the authors cite MuJoCo as an example of environments that don't contain excessive information, given that this is what they evaluate on. Overall, the reviewer raised a few excellent points on this issue, which in my opinion were not adequately addressed by the authors.
- W2 and W3 (The gating mechanism relies on a feature space that is split between either relevant or irrelevant features and extension to visual domains). The issue of context-dependent irrelevance is an important point raised by a number of reviewers, and the authors have not properly addressed it. They seem to suggest that there are enough environments where context-independent irrelevant features is an issue, yet they are mostly running on artificial environments that did not have this issue, or on toy experiments. As such, this unresolved concern indicates that either the paper is improperly scoped (i.e. they should limit their contributions to environments where context-independent irrelevant features may exist), or propose different architectural biases that can deal with context-dependent irrelevant features, so as to support the paper's main thesis more convicingly.

## JjLf
- W1 (The proposed architecture seems to be mainly applicable to settings with factored state spaces). Not properly addressed by authors. See discussion on context-independence above.
- W3 (we suggest comparing to other approaches designed for factorized state spaces like the CBM). Rather than running the proposed experiments, the authors (unconvincingly) argue for why the experiment is not meaningful.

**Reviewer Scores:**

- **QYMo:** Currently at 4, unlikely to increase given that concerns were not properly addressed.
- **q333:** Currently at 4, unlikely to increase given that concerns were not properly addressed.
- **RBxV:** Currently at 4, unlikely to increase given that concerns were not properly addressed.
- **JjLf:** Currently at 4, unlikely to increase given that concerns were not properly addressed.

---

### Decision · Program_Chairs · 2026-01-26

Reject